# Evaluating the influence of a constraint manipulation on technical, tactical and physical athlete behaviour

**Ben Teune**[1,2]*, **Carl Woods**[1], **Alice Sweeting**[1], **Mathew Inness**[1,2], **Sam Robertson**[1]

1 Institute for Health and Sport (iHeS), Victoria University, Melbourne, Australia, 2 Western Bulldogs, Melbourne, Australia

☯ These authors contributed equally to this work.
* benteune@outlook.com

## Abstract

Evaluating practice design is an important component of supporting skill acquisition and improving team-sport performance. Constraint manipulations, including creating a numerical advantage or disadvantage during training, may be implemented by coaches to influence aspects of player or team behaviour. This study presents methods to evaluate the interaction between technical, tactical and physical behaviours of professional Australian Football players during numerical advantage and disadvantage conditions within a small-sided game. During each repetition of the game, team behaviour was manually annotated to determine: *repetition duration*, *disposal speed*, *total disposals*, *efficiency*, and *disposal type*. Global Positioning System devices were used to quantify tactical (*surface area*) and physical (*velocity* and *high intensity running*) variables. A rule association and classification tree analysis were undertaken. The top five rules for each constraint manipulation had confidence levels between 73.3% and 100%, which identified the most frequent behaviour interactions. Specifically, four advantage rules involved high surface area and medium high intensity running indicating the attacking team's frequent movement solution within this constraint. The classification tree included three behaviour metrics: surface area, velocity 1SD and repetition duration, and identified two unique movement solutions for each constraint manipulation. These results may inform if player behaviour is achieving the desired outcomes of a constraint manipulation, which could help practitioners determine the efficacy of a training task. Further, critical constraint values provided by the models may guide practitioners in their ongoing constraint manipulations to facilitate skill acquisition. Sport practitioners can adapt these methods to evaluate constraint manipulations and inform practice design.

## Introduction

Sport coaches can design practice tasks to facilitate athlete development and support athlete learning and performance [1]. Coaches, along with other sport practitioners, should therefore consider the design of practice tasks which most effectively achieve their goals, whilst

**Data Availability Statement:** All relevant data are within the paper and its Supporting information files.

**Funding:** The author(s) received no specific funding for this work.

**Competing interests:** The authors have declared that no competing interests exist.

facilitating skill acquisition [2, 3]. A pedagogical approach, which may be used by practitioners to support the design of practice tasks, is constraint manipulation [2, 4]. Constraints represent boundaries or limitations to an athlete's interactions with their environment and the task being performed [5]. Constraint manipulations have been effective at guiding movement exploration and enhancing skill development in baseball batting [6] and swimming [7]. Specifically, in team sports, constraints such as field size or task rules, may be modified to guide the intentions, perceptions and actions of athletes while performing a practice task [8]. Athletes, therefore, must adapt their tactical (e.g. spatiotemporal movements), physical (e.g. distance and speed of locomotion), and/or technical (e.g. ball passing movements) behaviours to form movement solutions which aim to satisfy the constraints of a given task [9].

Evaluating the influence of a constraint manipulation on athlete behaviour is useful to understand the efficacy of what was manipulated and potentially support practitioners in (re) designing practice tasks [10]. The effect of constraint manipulations, including field size [11–13], the number of players [14–16] and task rules [15, 17, 18], on multiple facets of team and athlete behaviour, have been examined. To exemplify, field size manipulations can influence the lateral and longitudinal team width of players on the same and opposing teams [18, 19]. Field size is also positively related to the physical output of players, such as total distance covered [11]. Conversely, field size can be negatively related to the frequency of some technical actions, such as tackles or passes, in Australian football (AF) and field hockey [11, 13]. For example, if field size increased, the number of technical actions by athletes may decline due to the larger area available for athletes to move within. In contrast, when field size is decreased, the number of technical actions may be increased due to athletes needing to dispose the ball in a smaller area available. However, the interactions between a wider range of player behaviours, including technical, tactical and physical attributes, when manipulating constraints in AF training remains to be explored. Given the multi-faceted nature of sports performance, sports analysis should consider how such behaviours may interact and influence one another [20].

The constraints-led approach is a conceptual framework which advocates for the manipulation of practice task features (e.g. team size) to facilitate skill development [1, 4]. According to the constraints-led approach, constraints do not act in isolation but interact with one another, often in a non-linear manner [2]. Therefore, the manipulation of one constraint may have a dynamic influence on other constraints, with its influence changing or developing in different directions and over time. Thus, a challenge for practitioners is to understand how the manipulation of a single constraint can impact the many facets of an athletes performance [21]. Accordingly, it is pertinent to measure constraint interaction in order to provide appropriate contextual information when evaluating player behaviour [22, 23]. Importantly, determining constraint interactions highlights how the expression of a constraint changes when considered alongside other constraints. Further, from an applied perspective, the constraints-led approach has been suggested as an appropriate framework to support inter- and multi-disciplinarity in high performance support teams [20, 24]. For example, evaluating the skill and physical output of athletes together, associated with constraints manipulation in practice tasks, can foster interaction and collaboration between high performance and sports coaching staff [25, 26]. This may occur by providing a single report for multiple staff to cooperate in designing appropriate training environments to target complex goals in a single drill or training session. To this end, methods which can support practitioners to evaluate constraint interaction may enhance their training design.

Multivariate analytical techniques are advantageous for understanding constraint interaction [22, 24]. Such techniques, including rule association or classification and regression trees, have been applied to evaluate AF match kicking [22, 27], goal kicking [28] and skilled actions during training activities [10]. The advantages of these analyses have been discussed regarding

the prevalence of constraints during AF goal kicking [28]. Specifically, their flexibility to suit various data types, while considering non-linear relationships, and their ease of interpretability are highlighted. The interpretability and flexibility of analytical outputs should be considered to suit the needs of coaches and facilitate practical implementation of findings. Accordingly, the application of these techniques to inform team sport training design may be beneficial. Methods which can inform training design may support practitioners' decision making by guiding their attention toward key constraint interactions [24, 29]. Thus, the current study aimed to demonstrate methods to evaluate the influence of a numerical constraint manipulation on the interaction between technical, tactical and physical player behaviour.

## Methodology

### Participants

Participants were a convenience sample of professional players from one AF club ($n = 41$, height = 187.7 ± 8 cm, mass = 84.4 ± 8.6 kg, age = 24.7 ± 3.8 years). All players were injury free at the time of participation. Ethics approval was obtained from the Victoria University Human Research Ethics Committee (application number: HRE20-138). Written consent was provided by the club to use de-identified data collected from the participants, as a regular procedure during practice.

### Data collection

Data were collected for a single training task repeated ($n = 69$) throughout the 2022 Australian Football League pre-season training period (November 2021 –February 2022). Team selection was quasi-randomised by coaching staff on each occasion to balance team skill level. The training task comprised a small-sided game involving two teams of players competing against each other on a field approximately 80 m x 60 m (approximately 25% of a competition size AF field). The aim of the task was to move the ball from one end of the field to the other, while the defending team aimed to oppose this ball movement. The task ended when a shot on goal or a turnover was achieved. A team number constraint was manipulated by coaches, across all repetitions, whereby one team of seven competed against a team of eight, providing each team with either a numerical advantage (plus one) or disadvantage (minus one). For context, the practice task provided approximately 320 m$^2$/player while AF competition fields provide approximately 540 m$^2$/player. At the halfway point during each training session, the conditions were swapped so that both teams experienced each numerical constraint manipulation, in attack and defence. Task repetitions were defined by the sequences of play during the training activities, beginning with the ball at one end of the field until completion with the ball at the opposite end. Accordingly, repetitions were collected for both the numerical advantage ($n = 32$) and the disadvantage ($n = 37$) conditions.

To collect data pertaining to the technical skill of the players, the training activities were filmed from a side-on and behind-the-goals perspective with a two-dimensional camera (Canon XA25/Canon XA20). The two angles were subsequently aligned after the session for manual annotation. Skill data were collected via notational analysis software (Hudl Sportscode v12.4.2) using the aligned vision. Each pass (or "disposal") was manually coded according to the type (kick or handball) and effectiveness (effective or ineffective). A kick or handball < 40 m, in which the intended target retained possession of the ball, or a kick > 40 m to a 50/50 contest or advantage to the attacking team, was deemed effective, in accordance with Champion Data (Melbourne, Pty Ltd), the commercial statistics provider for the Australian Football League. A single coder notated this information. Thus, intra-rater reliability was examined via the kappa statistic [30], with a 14 day intra-reliability test resulting in "almost perfect"

**Table 1. Player behaviour metrics and associated definitions.** 1SD = one standard deviation.

| Type | Metric | Definition |
|---|---|---|
| Technical | Efficiency (%) | Percentage of effective disposals to total disposals |
| | Percentage Kicks (%) | Percentage of kicks to total disposals |
| | Total disposals (#) | Total number of disposals performed |
| | Repetition duration (s) | Time from beginning to end of repetition |
| | Disposal speed (disp/min) | Total disposals divided by repetition duration in minutes |
| Tactical | Surface Area ($m^2$) | Average surface area of attacking team minus average surface area of defending team |
| | 1SD Surface Area ($m^2$) | Standard deviation of surface area of attacking team minus standard deviation of surface area of defending team |
| Physical | Velocity (m/min) | Average velocity of attacking team minus average velocity of defending team |
| | 1SD Velocity (m/min) | Standard deviation of velocity of attacking team minus standard deviation of velocity of defending team |
| | HIR (m/min) | Average HIR metres per minute of attacking team minus average HIR metres per minute of defending team |
| | 1SD HIR (m/min) | Standard deviation of HIR of attacking team minus standard deviation of HIR of defending team |

agreement (0.95). Using this information, the efficiency, percentage of kicks, disposal count and disposal speed were calculated for each repetition (Table 1).

To determine tactical and physical movement of players during the training tasks, spatio-temporal positioning and velocity of each participant was collected using 10 Hz Global Positioning System devices (Vector S7, Catapult, Catapult Sports Ltd, Melbourne) which were placed on the participant's back, between their shoulder blades. Each participant wore the same device between sessions and during all activities to reduce inter-unit error. After session completion, tracking data for each participant was downloaded using the associated software (Openfield v 3.3.1) and exported for analysis. This data comprised latitude, longitude and velocity values at each 10 Hz timestamp for each participant. Each participant's tracking data was then down sampled to a rate of 1 Hz by taking the mean latitude, longitude, and velocity across every ten fixed samples. This was done to simplify the subsequent merging process with skill event data. This, and all subsequent data analysis, was completed using the *R* programming language [31] with the *RStudio* software (v2021.09.2).

Participant spatiotemporal data then was used to determine the surface area of each team during each task repetition. All latitude and longitude data were first converted to x and y coordinates, in metres, relative to the minimum x and y values in the dataset. Surface area was then calculated by determining the area ($m^2$) between the outermost players, at each 1 Hz time point, through the application of a convex hull [32]. For each repetition, the mean and one standard deviation (1SD) of the surface area was determined for the attacking and defending team. 1SD is a measure of the variation or dispersion of sample values relative to the mean. The mean and 1SD were then converted to a differential between the attacking and defending team. These calculations were performed to provide values which describe the attacking team's tactical movement relative to the defensive team.

The tracking data was also used to determine the velocity and high intensity running (HIR) metres of each team during each repetition. HIR was defined as any running speed $> 250$ m•min$^{-1}$ (or $>15$ km/h). The mean velocity was calculated for each player during each repetition and represented as m•min$^{-1}$. These values were then used to determine the mean and 1SD

in velocity for the attacking and defending team during each repetition. Similarly, HIR was calculated for each repetition and mean HIR was calculated for each player during each repetition and represented as m•min$^{-1}$. These values were then used to determine the mean and 1SD in HIR for each team during each repetition. Mean velocity, velocity 1SD, mean HIR, and HIR 1SD were represented as a differential between the attacking and defending team to provide values for the attacking team's physical movement relative to the defence.

### Statistical analysis

A correlogram was used to explore any univariate linear relationships between the behaviour metrics, as listed in Table 1. To determine the influence of the team number constraint manipulation on player behaviours, two multivariate analytical approaches were applied: rule association and classification trees. To apply rule association, each behaviour metric was first discretised into three arbitrary categories: low, medium and high. These categories were chosen to align with the preferred output style of the end-users (i.e., coaches of the football club). This was achieved using the *discretizeDF* function in the *arules* package [33], using a cluster method set for three groups. Rules for each numerical condition were then generated using the *apriori* function, which uses the *Apriori* algorithm [34]. The *Apriori* algorithm identifies relationships between variables by producing rule sets, similar to if-then statements. For example, the rule {Efficiency = x, Surface Area = y} = > {Velocity = z} indicates if antecedent values of Efficiency and Surface Area occurred, then the consequent value of Velocity occurred. Rules may be evaluated via support (%), the frequency of a rule within a dataset, and confidence (%), the frequency of the consequent given the antecedents of the rule. Parameters of the *apriori* function were set to search for rules with a minimum support of 0.15, minimum confidence of 0.7, and a minimum rule length of four.

The second approach applied a classification tree using the *rpart* package [35]. The *rpart* function was used to classify the constraint condition of each task repetition based on the values of the behaviour metrics. The *rpart* function achieves this by partitioning the data according to specific values of variables which are most strongly linked to the outcome variable. The default parameters for the function were used with a complexity parameter of 0.01, a minimum split attempt of 29% (20 observations) and minimum terminal node observations set at seven (minimum split / 3).

## Results

For the 32 numerical advantage repetitions, the mean duration was 16.3 s ± 8.2 s and the mean disposal count was 2.9 ± 1.3. For the 37 numerical disadvantage repetitions, the mean duration was 22.7 s ± 12.8 s and the mean disposal count was 3.6 ± 1.6. The distribution of each metric, within each condition is displayed in Fig 1. The correlogram was presented in Fig 2. Univariate correlations between all behaviour metrics were within 0.5 and -0.5 with the exception of positive correlations between total disposals and repetition duration (0.84) and between velocity and HIR (0.8).

For the rule association approach, the resulting cut-off values used during discretisation are displayed in Table 2 and the counts within each category of the discretisation are displayed in Fig 3. From the results of the *Apriori* algorithm, nine rules were generated for the numerical advantage condition and six rules were generated for the numerical disadvantage condition. The top five rules, by confidence, for each condition are displayed in Figs 4 and 5. For the numerical advantage condition, confidence ranged from 80% to 100% and for the numerical disadvantage condition, confidence ranged from 73.3% to 85.7%.

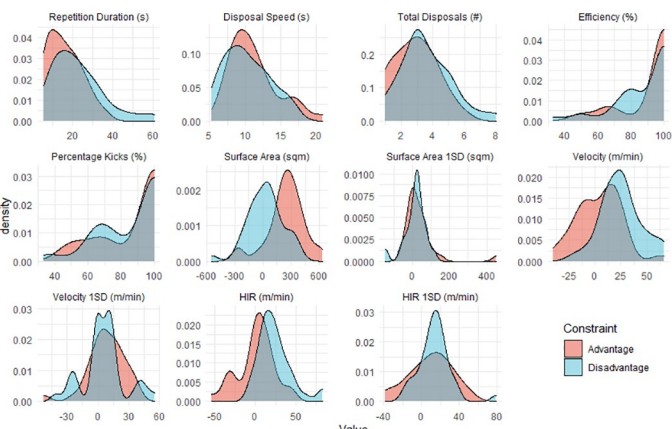

**Fig 1. Distribution of each behaviour metric within advantage (red) and disadvantage (blue) constraint conditions.**

The resulting model for the classification tree is displayed in Fig 6. The only variables used by the model to partition the data were surface area, repetition duration and velocity 1SD. Four terminal nodes are shown, two for each numerical condition with classification accuracies ranging from 71% to 94%. A visualisation of all behaviour metrics within each terminal node, scaled to allow comparison, was also provided (Fig 7).

## Discussion

The aim of this study was to demonstrate methods to evaluate a numerical constraint manipulation while considering the interaction of player technical, tactical and physical behaviour. A rule association and classification tree approach were used to analyse player behaviour, under

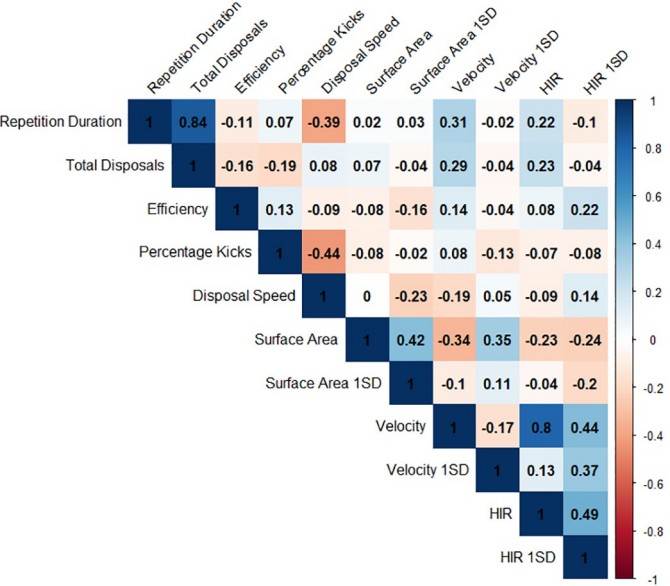

**Fig 2. Correlogram of each behaviour metric.** Each tile is labelled with the correlation coefficients between each metric and coloured according to this value as per the colour scale on the right (blue hues indicate a positive correlation and red hues indicate negative correlation).

**Table 2. Cut-off values used to discretise each behaviour metric.**

| Metric | Low | Med | High |
|---|---|---|---|
| **Repetition Duration (s)** | < 18.3 | 18.3 to 38.2 | > 38.2 |
| **Total Disposals (#)** | < 2.29 | 2.29 to 3.89 | > 3.89 |
| **Disposal Speed (disp/min)** | < 10 | 10 to 14.2 | > 14.2 |
| **Efficiency (%)** | < 61.3 | 61.3 to 88 | > 88 |
| **Percentage Kicks (%)** | < 69.3 | 69.3 to 88.8 | > 88.8 |
| **Surface Area (m²)** | < -28.3 | -28.3 to 237 | > 237 |
| **Surface Area 1SD (m²)** | < 11.7 | 11.7 to 250 | > 250 |
| **Velocity (m/min)** | < 3.61 | 3.61 to 36.7 | > 36.7 |
| **Velocity 1SD (m/min)** | < -8.95 | -8.95 to 21.5 | > 21.5 |
| **HIR (m/min)** | < -11.7 | -11.7 to 27.1 | > 27.1 |
| **HIR 1SD (m/min)** | < 0.46 | 0.46 to 27.2 | > 27.2 |

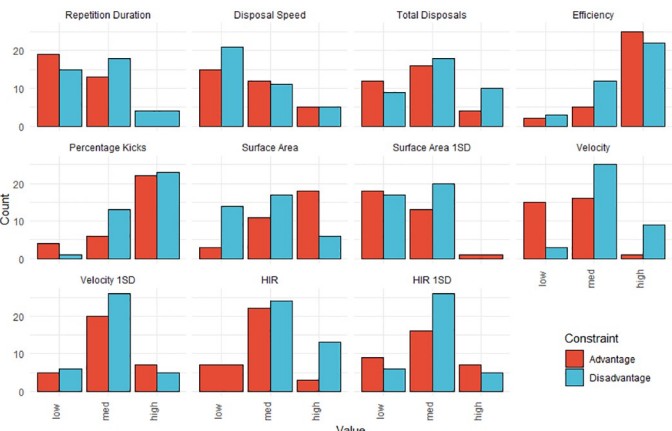

**Fig 3. Results of the discretisation of each behaviour metric.** Repetition counts for each category are displayed for the advantage (red) and disadvantage (blue) constraint conditions.

the premise of supporting the design of practice tasks in team sport. The rule association provided a simple visualisation whereby coaches can identify associations between aspects of player behaviour. Additionally, the classification tree could be used to determine specific values of interest which can guide ongoing constraint manipulations in practice task designs.

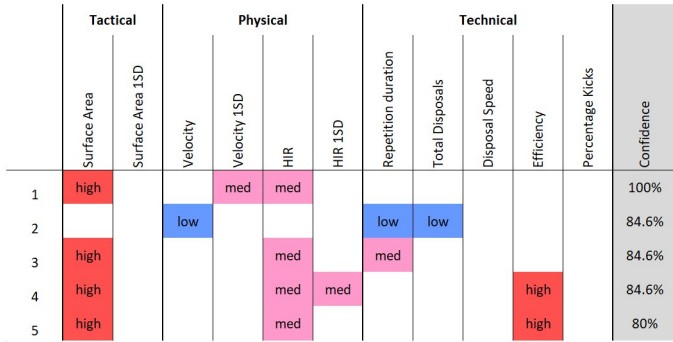

**Fig 4. The top five rules generated for the advantage constraint condition, ordered by confidence.** Each discretised metric is colour coded according to its category (red = high, pink = med, blue = low) for visual interpretability.

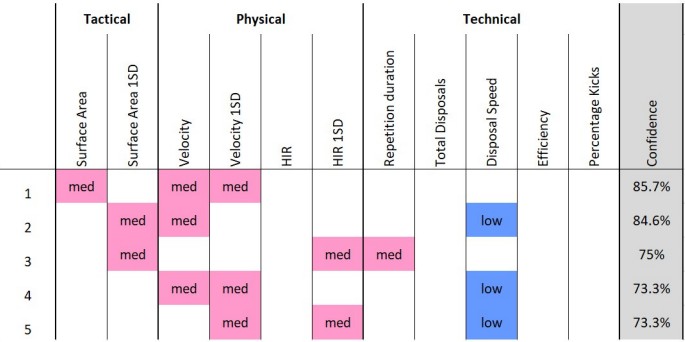

**Fig 5. The top five rules generated for the disadvantage constraint condition, ordered by confidence.** Each discretised metric is colour coded according to its category (red = high, pink = med, blue = low) for visual interpretability.

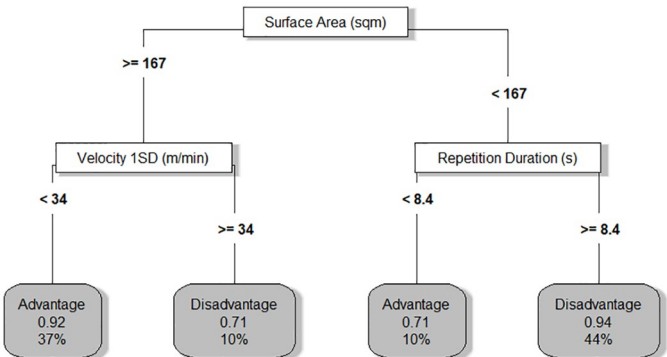

**Fig 6. The classification tree used to model the constraint condition (advantage or disadvantage).** Terminal nodes are labelled with the predicted constraint condition while the decimals indicate the accuracy of the fitted value and the percentages indicate the frequency of observations.

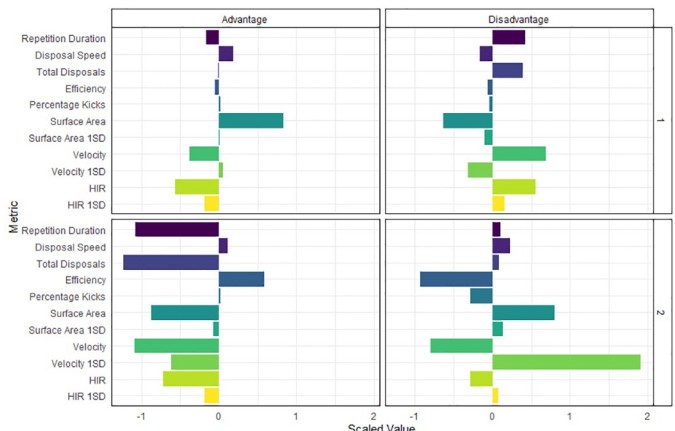

**Fig 7. The average of each behaviour metric within the identified task solutions (1 and 2) for each constraint condition (advantage and disadvantage).** The bar plot values are scaled to a mean of zero and a standard deviation of one to allow comparability between metrics.

The results of the rule association analysis provide a simple heuristic which could support coach decision-making. The rules displayed in Figs 4 and 5 highlight which simultaneous behaviours players are exploiting to achieve the given task. This builds upon previous AF work using rule association to evaluate training [10] and match play [22, 27] through the inclusion of tactical and physical behavioural metrics. Moreover, the rule association identified non-linear relationships between behaviour metrics which were not determined in the linear exploration shown in Fig 2. Discretising continuous variables is a necessary step to perform rule association and presents both advantages and disadvantages for interpretation. Binning values into three categories; low, medium and high, may suit the communication preferences of coaches although, other quantities of bins may also be used. Decisions on bin quantities should be aimed at improving the coaches' ease of use and increasing the speed of their decision making, which therefore may vary. However, discretisation can reduce the explanatory power of continuous variables. For example, a range of values can be identified within each category but no specific values for player behaviour can be provided to the practitioner, limiting their utility for intervention.

The results of the rule association suggest that, when playing with a numerical advantage, teams used their additional player to spread over larger areas than their opposition. This was indicated as four of the five top rules for the advantage condition included high levels of surface area. Additionally, within each of these four rules, high surface area was associated with medium levels of HIR. This suggests that this level of physical running speed was required to achieve the levels of high surface area. Other metrics, including kick percentage and disposal speed, were not included in any of the top five rules. This indicated that the numerical advantage did not influence these behaviours, nor did they interact with others at a meaningful level. Contrastingly, in the numerical disadvantage condition, three of the top five rules involved low disposal speed. A team at disadvantage frequently exhibited a slower speed of play. Low disposal speed was also associated with medium surface area 1SD, medium velocity and medium velocity 1SD. Similar findings in investigations of other constraint manipulations, such as field density or team size, have reported simultaneous changes to skilled, physical and tactical behaviour of players in field hockey and soccer [13, 36] however, their interactions were not determined. In the current study, results of the rule association showed how interactions between the behaviours of players can be measured. Accordingly, these interactions are pertinent information for both a conditioning and skills coach. For example, a conditioning coach can monitor and prepare players for the specific work rates required to perform tactical manoeuvres influenced by the numerical constraint manipulation. This outcome highlights how the analysis can provide a platform for a multidisciplinary approach to support athlete development [24, 37].

The second rule for the numerical advantage condition presented three unique variables which were absent in any other rules. These variables were low repetition duration, low total disposals and low velocity. This indicates an alternate task solution was used by the players. In this solution, the ball is moved quickly down the field with a low quantity of disposals and lower running speed than the defence. This observation is similar to other work in AF, in which the inclusion of an additional attacker reduced the average velocity of the group [14]. This solution may emerge given a sudden exploitation of an opportunity, such as a lapse in defensive structure. Depending on the training objectives of coaches, training design may be modified to encourage or discourage performance of this solution. For example, to discourage this solution and further guide player's attention toward using their numerical advantage to maximise surface area, an additional task constraint of a minimum pass count could be implemented during the advantage condition.

Contrasted with rule association, the classification tree could be advantageous by enabling the data to be modelled in its continuous format. Accordingly, when using numerical data, critical values can be directly provided by the model which are influential on player behaviour. To

exemplify, along the right branch of the tree (Fig 6), a common task solution for the numerically disadvantaged team was to slow the sequence of play down as indicated by the repetition duration of >8.4 s. This behaviour may have emerged as players sought additional time to create space against a team possessing an extra number, thereby maintaining possession of the ball. The repetition duration value of 8.4 s may be leveraged by a coach seeking to encourage greater exploration in task solutions. For example, a temporal constraint of 8 s may be introduced to challenge the stability of this solution for the team with the numerical inferiority. This may lead to the emergence of a new behavioural pattern, as players search to exploit both the numerical inequality and temporal constraint. Only three behaviours were found to be influenced by manipulation of the numerical constraint: surface area, velocity 1SD and repetition duration. This suggested that all other behaviours remained predominantly stable despite the numerical constraint manipulation. Using this information, coaches may choose to manipulate additional constraints, such as field dimensions or task rules, to perturb player behaviours and encourage variability [38].

The partitions provided by the classification tree may be used to identify the different task solutions performed by teams within each numerical constraint. A similar approach has been reported in swimming where a clustering analysis identified if learners were exploiting or exploring task solutions during training [7]. In the current study, the classification tree produced two terminal nodes for each numerical condition, suggesting two unique task solutions were exhibited within each constraint. The first solution was the most frequently used (advantage = 37%, disadvantage = 44%) and the second solution was the least frequently used (advantage = 10%, disadvantage = 10%). Fig 7 can thus highlight how technical, tactical and physical behaviours are organised simultaneously by teams to achieve the task goal. This may be advantageous as a complementary visualisation to the classification tree, reporting all behaviour metrics in addition to the three included in the classification tree. Through evaluations of these behaviours, coaches may seek to guide or nudge players towards new or more optimal task solutions, according to their training objectives [3].

Given the applied nature of the current study, some limitations exist which should be considered. Field sizes were approximately measured during data collection and some small variations may exist between training sessions. This, however, was controlled as closely as practically possible. Additionally, while players on each team were selected to balance skill level, player selection was inconsistent across each session. Accordingly, these factors may have influenced team behaviours between task repetitions. Some instances occurred where there was an unused player on the sideline (due to irregular numerical grouping) and players were permitted to substitute between repetitions. A total of 16 substitutions occurred during data collection which may have influenced the physical output of players. Although the validity and reliability of 10 Hz Global Positioning Systems have been assessed [39, 40], mean error of 96cm has been shown in such units [41]. It is unlikely this margin of error will have influenced results, given the large field sizes used, however this is yet to be determined. From an analytical perspective, only one measure of tactical behaviour was used during this study and future work may be directed to include other measures of collective team behaviour, such as centroid location, difference between team centroids, or team separateness. Finally, future work may seek to measure constraints on disposals, such as pressure or possession time, to provide further context to the technical actions performed during repetitions. The results, nonetheless, provide an enticing methodological platform for future work.

## Conclusion

This study applied two multivariate analytical techniques, rule association and a classification tree, to evaluate the influence of a numerical advantage or disadvantage on the technical,

tactical and physical behaviour of AF players during a small-sided training task. The rule association approach presented a simple and interpretable output for coaches which informed interactions between key behaviours during each constraint condition. The classification tree provided specific values of interest which may be used to inform further constraint manipulations to enhance practice task design. A visualisation of the different task solutions identified through the classification tree was provided to assist coaches in evaluating how players organise their movements within each constraint. These methods and visualisations are provided as tools which sport practitioners are encouraged to adopt to inform the design of their own training activities.

## Supporting information

**S1 Data. Advantage and disadvantage task repetitions.**
(CSV)

## Author Contributions

**Conceptualization:** Ben Teune, Sam Robertson.

**Data curation:** Ben Teune.

**Formal analysis:** Ben Teune.

**Methodology:** Ben Teune, Carl Woods, Sam Robertson.

**Supervision:** Carl Woods, Alice Sweeting, Mathew Inness, Sam Robertson.

**Visualization:** Ben Teune.

**Writing – original draft:** Ben Teune.

**Writing – review & editing:** Carl Woods, Alice Sweeting, Mathew Inness, Sam Robertson.

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
