## [Decision Letter · Decision Letter 0]

7 Sep 2022

PONE-D-22-19072Evaluating the influence of a constraint manipulation on technical, tactical and physical athlete behaviourPLOS ONE

Dear Dr. Teune,

Thank you for submitting your manuscript to PLOS ONE. After careful consideration, we feel that it has merit but does not fully meet PLOS ONE’s publication criteria as it currently stands. Therefore, we invite you to submit a revised version of the manuscript that addresses the points raised during the review process.

While minor revisions are required, greater clarification is necessary throughout the paper to clearly convey the the approach undertaken and further justify your conclusions ==============================

We look forward to receiving your revised manuscript.

Kind regards,

Chris Connaboy

Academic Editor

PLOS ONE

Journal Requirements:

Reviewers' comments:

Reviewer's Responses to Questions

**Comments to the Author**

1. Is the manuscript technically sound, and do the data support the conclusions?

Reviewer #1: Yes

Reviewer #2: Yes

2. Has the statistical analysis been performed appropriately and rigorously? 

Reviewer #1: Yes

Reviewer #2: Yes

3. Have the authors made all data underlying the findings in their manuscript fully available?

Reviewer #1: Yes

Reviewer #2: Yes

4. Is the manuscript presented in an intelligible fashion and written in standard English?

Reviewer #1: Yes

Reviewer #2: Yes

5. Review Comments to the Author

Reviewer #1: The current manuscript details a study the presents a method to evaluate the effect of a constraint manipulation, specifically a numerical advantage or disadvantage, on key behaviours of professional footballers during a small-sided game.

In my opinion the manuscript is well-written and easy to follow and examines an area of scientific enquiry that is much needed – that is providing measures and analysis techniques that can explain the impact constraint manipulations have on behaviour. The authors should also be commended on the ability to utilise elite athletes with reasonably high numbers, something that is often difficult to achieve in practice (notwithstanding some of the limitations the authors have highlighted).

I have a few points that I believe are needed to be addressed before I could recommend this article for publication which mainly focus on the method and the level of detail provided.

My first main point is the use of the GPS latitude and longitude data for spatiotemporal measures e.g., surface area. As far as I can tell, there is no indication in the manuscript about the possible error in reporting for absolute position measured by GPS devices (which is then used to calculate surface area). I recognise the difficulties in being able to determine these types of measures but in previous literature e.g., Linke et al (2018) there was a reported average measurement error of 96cm. I appreciate the technology has improved since then and this study uses different devices, but I would suggest that the authors either provide some information about the potential error in the measurement or at least acknowledge the potential for this to impact the findings.

The next points relate to several parts of the methodology that I believe requires clarification.

Page 5, Line 92 – how does the size of the area (85m x 65m) compare to a regular AFL field (understanding there are differences) or perhaps the regular training field? I think this information is useful in understanding the amount of space available in the small-sided game relative to a real match or training session

Page 5, Line 92-93: “aim of the task was to move the ball…” – can the authors please provide additional information about how the task ends i.e., is the end goal to maintain possession in a dedicated “goal” or just get the ball (regardless of possession) to a target i.e., like kicking a goal during a match

Alongside the above comment, it is not clear to me what happens if there is a turnover of possession. If there is a turnover does that end the trial or does the defending team become the attacking team?

Page 5, Line 94: “seven competed against a team of eight” – Is it possible to express the number of players per metre squared, in a similar way to Oppici et al. (2018)? This could also assist the earlier comment about the size of the field compared to normal match scenario.

Page 7, Line 124-125 – regarding the down sampling of data to 1Hz, the reason provided by the authors is understandable as I presume this is to align the data with the video (although the Hz of the video footage is not provided). If the video was 25Hz, could the authors have chosen to down sample to 5Hz instead? Are there any potential concerns by down sampling to 1Hz about the positional data reported?

The final point relates to the interpretation that the rule association analysis suggested that teams at a numerical advantage used their additional player to spread over larger areas than their opposition. The evidence provided for this is that four of the five top rules included high levels of surface area. I may be misinterpreting the analysis here, but to my understanding the surface area measurement is the difference between the attacking and the defending team (i.e., “average surface area of attacking team minus average surface area of defending team”, Table 1). If so, is it not also possible that this difference is found because the defending team “shrinks” their space (i.e., lowers their average surface area)?

The above raises more questions for me about the decision to use attacking team minus defending team for the tactical and physical variables, but not for the technical. For example, is there a reason why simply the average surface area of the attacking team was not used directly? I am not suggesting the analysis needs to be re-run, but I would encourage the authors to provide further information to justify the use of the differential values.

Minor points

Page 3, Line 49-50: “field size manipulations can influence…” – I would suggest adding to this further by providing a specific explanation on how field size manipulations influence behaviour.

Page 4, Line 64: “skilled” – this appears to be a typo, perhaps should be “skill”?

Page 14, Line 286: “permitted to substitute between repetitions” – not a major point but is it possible to report the number of substitutions made in some way?

Comment

This is just a comment, but I really like the point made on Page 13, Lines 259-261 which clearly show the practical application of this type of analysis.

References

Luca Oppici, Derek Panchuk, Fabio Rubens Serpiello & Damian Farrow (2018) Futsal task constraints promote transfer of passing skill to soccer task constraints, European Journal of Sport Science, 18:7, 947-954, DOI: 10.1080/17461391.2018.1467490

Linke D, Link D, Lames M (2018) Validation of electronic performance and tracking systems EPTS under field conditions. PLOS ONE 13(7): e0199519. https://doi.org/10.1371/journal.pone.0199519

Reviewer #2: In general this is a well put together study, proposing nice summative solutions to an applied problem. There are some minor clarifications and details needed throughout, as detailed below, but all in all I think this is a worthy body of research.

6. PLOS authors have the option to publish the peer review history of their article (what does this mean?). If published, this will include your full peer review and any attached files.

Reviewer #1: No

Reviewer #2: No

---

## [Author Response · Author response to Decision Letter 0]

10 Oct 2022

Reviewer #1

Reviewer #1: The current manuscript details a study the presents a method to evaluate the effect of a constraint manipulation, specifically a numerical advantage or disadvantage, on key behaviours of professional footballers during a small-sided game.

In my opinion the manuscript is well-written and easy to follow and examines an area of scientific enquiry that is much needed – that is providing measures and analysis techniques that can explain the impact constraint manipulations have on behaviour. The authors should also be commended on the ability to utilise elite athletes with reasonably high numbers, something that is often difficult to achieve in practice (notwithstanding some of the limitations the authors have highlighted).

• Response: We would first like to thank the reviewer for their time and expertise in reviewing this manuscript. Please find our responses to your comments below and where necessary, manuscript amendments in blue font colour.

I have a few points that I believe are needed to be addressed before I could recommend this article for publication which mainly focus on the method and the level of detail provided.

My first main point is the use of the GPS latitude and longitude data for spatiotemporal measures e.g., surface area. As far as I can tell, there is no indication in the manuscript about the possible error in reporting for absolute position measured by GPS devices (which is then used to calculate surface area). I recognise the difficulties in being able to determine these types of measures but in previous literature e.g., Linke et al (2018) there was a reported average measurement error of 96cm. I appreciate the technology has improved since then and this study uses different devices, but I would suggest that the authors either provide some information about the potential error in the measurement or at least acknowledge the potential for this to impact the findings.

• Response: Although the exact margin of error for the devices in the current study are unknown, the potential for this to influence results has now been acknowledged (lines 316-319)

The next points relate to several parts of the methodology that I believe requires clarification.

Page 5, Line 92 – how does the size of the area (85m x 65m) compare to a regular AFL field (understanding there are differences) or perhaps the regular training field? I think this information is useful in understanding the amount of space available in the small-sided game relative to a real match or training session

• Response: The authors recognise this would be beneficial information for readers unfamiliar with Australian Football and has now been provided (line 105)

Page 5, Line 92-93: “aim of the task was to move the ball…” – can the authors please provide additional information about how the task ends i.e., is the end goal to maintain possession in a dedicated “goal” or just get the ball (regardless of possession) to a target i.e., like kicking a goal during a match

• Response: The task ended with a shot on goal or a turnover. This detail has now been included (line 107).

Alongside the above comment, it is not clear to me what happens if there is a turnover of possession. If there is a turnover does that end the trial or does the defending team become the attacking team?

• Response: The trial ended with a turnover. This detail is now included at line 107.

Page 5, Line 94: “seven competed against a team of eight” – Is it possible to express the number of players per metre squared, in a similar way to Oppici et al. (2018)? This could also assist the earlier comment about the size of the field compared to normal match scenario.

• Response: This information has now been included to provided greater context on the space provided to players (line 110-111)

Page 7, Line 124-125 – regarding the down sampling of data to 1Hz, the reason provided by the authors is understandable as I presume this is to align the data with the video (although the Hz of the video footage is not provided). If the video was 25Hz, could the authors have chosen to down sample to 5Hz instead? Are there any potential concerns by down sampling to 1Hz about the positional data reported?

• Response: We don’t believe there are any concerns using down sampled 1Hz data. The outcome measures which were derived from the tracking data were summarised across multiple players and across the duration of each repetition (mean surface area, mean velocity and mean HIR). Given the analysis didn’t account for any continuous temporal information, we don’t believe the resolution of the data, 1Hz or 5Hz, would affect these measures greatly.

The final point relates to the interpretation that the rule association analysis suggested that teams at a numerical advantage used their additional player to spread over larger areas than their opposition. The evidence provided for this is that four of the five top rules included high levels of surface area. I may be misinterpreting the analysis here, but to my understanding the surface area measurement is the difference between the attacking and the defending team (i.e., “average surface area of attacking team minus average surface area of defending team”, Table 1). If so, is it not also possible that this difference is found because the defending team “shrinks” their space (i.e., lowers their average surface area)?

• Response: Regardless of the perspective of “shrinking” or “expanding”, the attacking team still position themselves over a larger surface area than the defending team (i.e. the attackers did not equalise the shrinking surface area of the defence). Because the behaviour metrics in this study were recorded relative to the attacking team, we expressed our discussion in alignment with this.

The above raises more questions for me about the decision to use attacking team minus defending team for the tactical and physical variables, but not for the technical. For example, is there a reason why simply the average surface area of the attacking team was not used directly? I am not suggesting the analysis needs to be re-run, but I would encourage the authors to provide further information to justify the use of the differential values.

• Response: To address the reviewers first point, given the absence of technical information for the defending team, no comparison could be drawn between attackers and defenders. Secondly, given the aim of the task was on ball movement, all behaviour metrics were summarised as relative to the attacking team (i.e. the team with the ball). In sport, the physical and tactical movement of attacking players is influenced by the movement of the defence. Therefore, it made more practical sense to calculate values which described the attacking teams movement relative to the defence. This detail has now been included at lines 150-151 and 160.

Minor points

Page 3, Line 49-50: “field size manipulations can influence…” – I would suggest adding to this further by providing a specific explanation on how field size manipulations influence behaviour

• Response: A more specific result has now been included here on how field size influences lateral and longitudinal team width (line 53).

Page 4, Line 64: “skilled” – this appears to be a typo, perhaps should be “skill”?

• Response: This has been amended (line 75).

Page 14, Line 286: “permitted to substitute between repetitions” – not a major point but is it possible to report the number of substitutions made in some way?

• Response: Across the 69 repetitions a total of 16 substitutions were made. This information has been included in the limitations section (line 315-316)

Comment

This is just a comment, but I really like the point made on Page 13, Lines 259-261 which clearly show the practical application of this type of analysis.

• Response: Thank you.

References

Luca Oppici, Derek Panchuk, Fabio Rubens Serpiello & Damian Farrow (2018) Futsal task constraints promote transfer of passing skill to soccer task constraints, European Journal of Sport Science, 18:7, 947-954, DOI: 10.1080/17461391.2018.1467490

Linke D, Link D, Lames M (2018) Validation of electronic performance and tracking systems EPTS under field conditions. PLOS ONE 13(7): e0199519. https://doi.org/10.1371/journal.pone.0199519

Reviewer #2: In general this is a well put together study, proposing nice summative solutions to an applied problem. There are some minor clarifications and details needed throughout, as detailed below, but all in all I think this is a worthy body of research.

• Response: Thank you again for your time and expertise reviewing our manuscript.

 

Reviewer #2

General Comments:

In general this is a well put together study, proposing nice summative solutions to an applied problem. There are some minor clarifications and details needed throughout, as detailed below, but all in all I think this is a worthy body of research.

• Response: We would first like to thank the reviewer for their time and expertise in reviewing this manuscript. Please find our responses to your comments below and where necessary, manuscript amendments in red font colour.

Comments to Authors:

Abstract:

Line 26: Please specify specific findings in results here.

• Response: More specific results, on the rules and team behaviour metrics, have been included in the abstract now (lines 26-29 and 32-33)

Introduction: 

Line 33: Practice can occur without a coach. As an opening sentence is so important, I would encourage tweaking the wording here to let the next sentence flow better. Something like ‘coaches can design practice sessions to facilitate athlete development.’

• Response: Amended as suggested (lines 37-38)

Line 46: Efficacy of what? Please restructure sentence. Something like ‘…to understand th efficacy of practice tasks, and potentially support practitioners in designing these.’

• Response: This sentence has now been clarified to show we mean the efficacy of constraint manipulations (line 50).

Line 50: Provide an example of physical output

• Response: Included as suggested (line 55).

Line 51: Related to the frequency how? Clarify if more or less for the reader.

• Response: We have included some clarification on the inverse relationship between field size and technical action frequency (i.e. the larger the field size, the less frequent the actions). Lines 57-60.

Line 53: As technical, tactical and physical attributes are key components running through this paper, these need to be defined and introduced, and the relationships between them explored a little more to provide a rationale for investigating these relationships. Whilst they appear logical, they need appropriate justifications for the reader.

• Response: In regards to the attribute definitions, we would draw the reviewers attention to lines 46-48 where examples of these attributes are provided. A sentence has also been added to this paragraph highlighting the multi-faceted and intertwined nature of sports performance to justify the relationships between technical, tactical and physical attributes of player performance. (Glazier, 2017) has also been included as a supporting reference here (line 62-63).

Line 55: Avoid the use of the word ‘constraints’ to define constraints. An alternative word is needed to define this in this sentence…not just a word repeat. This becomes particularly important with the repeat use of the word ‘constraint’ throughout the following paragraph.

• Response: The term “constraints” has been replaced with “practice task features” to clarify this sentence. (line 65)

Line 57: This needs explaining.

• Response: Further explanation on the non-linear interaction of constraints has been included (line 67-68).

Line 59: This also needs clarifying.

• Response: The sentence following this was intended to elaborate on the importance of measuring constraint interaction. This sentence has now been amended to link these two more effectively (line 72).

Line 61: What do you mean by ‘contextualising constraint interactions’. Try to avoid ambiguous phrases such as this, and support the reader in following your narrative.

• Response: This comment relates to the above. Accordingly, “contextualising” has now been replaced with “determining” to improve the clarity for the reader (line 72).

Line 64-66: How does this foster interaction and collaboration? Please create a stronger link here to underpin your core arguments.

• Response: An additional sentence has been included to explain this connection in greater detail (line 77-79)

Line 71-72: Your explanation of multivariate techniques is good, however rather than just referring the reader to a paper that has demonstrated advantages, please add a sentence to explicitly clarify what these are and how you can us them in your context. This way, the reader does not need to await for the methods if they are not familiar.

• Response: a sentence has been added briefly describing the advantages of multivariate techniques which are discussed in the referred paper (line 85-86) 

Methodology:

Line 86: Should this be ‘as a regular procedure…’? E.g. grammar.

• Response: This has now been amended (line 99).

Line 91-92: Adding some words here to qualify the size of the field compared to a full AF oval would help the non-AF readers here (e.g. reduced field of play).

• Response: This detail has now been included to clarify for non-AF readers (line 105).

Line 100: Assume in total over the n=69? Perhaps add here to re-clarify.

• Response: This is now clarified (line 108). The breakdown of advantage and disadvantage repetitions is also provided at line 115-116.

Table 1: Good level of detail!

• Response: Thank you.

Line 119-120: Was the same unit wore in each session too, or just within each session?

• Response: The same unit was worm between sessions too. This is clarified in the manuscript now at line 135.

Line 122-124: Why was this meaned, rather than taking the largest value every 10 Hz? Meaning the data would down sample and artificially smooth to be lower. So effectively, the velocity value obtained would always be lower than reality (and not-systematically). I think it could be worth quickly checking the data to see if it makes much difference. 

• Response: To check this we examined a single player’s data from a single session. We compared the max smoothing method, as per the reviewers comment with the mean smoothing, as per the original manuscript methods. The mean smoothing provided a closer representation of the original data so we believe this to be the most useful method.

Line 130-131: This is creating and applying a convex hull, rather than ‘being known as one’. Please rephrase to correctly state this detail.

• Response: This detail is now amended (line 146).

Line 136: 1SD needs defining (not just as an acronym) 

• Response: A sentence defining 1SD as a measure of variation or dispersion has been included (line 148-149).

Line 138: Perhaps move definition (‘>250m.min-1’) earlier. 

• Response: Definition now moved to the sentence following the first mention of HIR (line 153-154).

Line 143: Please explicitly state behaviour metrics here.

• Response: A reference to Table 1 has been included here, which lists and defines each of the behaviour metrics (line 163).

Line 147: Please clarify what ‘preferences of the end-users’ means.

• Response: A clarification on this has been included at line 166.

Line 150: Please state what the Apriori algorithm does in brief) rather than just refer to another paper. Your paper should be standalone in its readability.

• Response: An explanation of the Apriori algorithm is provided (lines 169-173)

Line 151-152: I am not familiar with this approach. Please can you define / explain / clarify what ‘each numerical condition’ is in the text and what these parameters mean.

• Response: The numerical condition refers to the constraint manipulation (numerical advantage or disadvantage) which has now been removed upon revision. Further explanation of the Apriori approach, and the associated parameters, is provided to explain for readers unfamiliar with this analysis (lines 173-175).

Line 154: Please briefly classify how rpart function does this (in a sentence).

• Response: A short explanation on the rpart function is provided (line 179-180).

Results:

Lines 170-174: I feel like more explanation was needed in the methodology to explain these results. This is exemplified by lines 173-174 where the authors are having to explain interpretation of results. A results section should not need explanations.

• Response: A detailed explanation of the rule association methods has now been provided in the methods section (line 173-175) and explanations removed from the results section.

Figure 1: Should density have a unit? If not, state in figure caption perhaps. If the author feels strongly about this, I am happy with as is.

• Response: The units for density are relative to the probability of each given metric. Thus, density units would be different in each panel. In this case, we have chosen to leave density without a unit measurement as it is the shape of the distribution which is more relevant to the paper, rather than the probability over the distribution.

Figure 2: More detail needed to explain graph in caption (e.g. colour legend needs labelling)

• Response: More explanation has been provided for this figure regarding the values and the colour scale (line 193-195)

All figures: Colour coding should be explained in figure captions in detail, not just a statement that colour coding exists.

• Response: Details regarding colour coding have now been included in the captions of all relevant figures.

Discussion:

Line 203-204: Perhaps consider rephrasing wording at the end of the sentence (e.g. ‘the design of practice tasks’ is more easily read).

• Response: This amendment is now included at line 232-233.

Lines 204-205: I feel that this statement may be a bit early, or a little strong, given that this has not been discussed or demonstrated yet. Please temper this statement, or address this later in the discussion when your argument has been presented and supported (e.g. a usual pattern is to present a statement from your findings, discuss and support, then link to existing literature).

• Response: The premise of this opening discussion paragraph is to provide a recap of aims and methods, and a short preview of the discussion, as a foreshadow of what will be discussed in greater detail throughout the remainder of the section.

Lines 205-207: As above. These statements need support.

• Response: As above, these statements are intended to be a preview of the main discussion points.

Line 222: Comma should be after the word ‘that’.

• Response: This correction has been made (line 251)

Line 241-243: This is a large sentence and feels a little clumsy. Please consider rewording.

• Response: This sentence has now been separated into two (line 270-271)

Line 251: Do you have a reference to support the effectiveness of this constraint? Just a thought if proposing changes to practice based on your findings.

• Response: Unfortunately, we are unaware of any literature which specifically investigates the use of this constraint. Although, this constraint example was drawn from practical experience, the intention of providing this example is to provoke thought on how ongoing constraint manipulations may be used to further guide the exploration of player behaviours. The specific constraint to be used is of less importance.

Limitations (lines 285-292): The work you have undertaken is great, practical, ecologically valid. I feel this should be applauded and stated in here somewhere to offset limitations.

• Response: We thank the reviewer for this comment and understanding in this work.

---

## [Decision Letter · Decision Letter 1]

21 Nov 2022

Evaluating the influence of a constraint manipulation on technical, tactical and physical athlete behaviour

PONE-D-22-19072R1

Dear Dr. Teune,

We’re pleased to inform you that your manuscript has been judged scientifically suitable for publication and will be formally accepted for publication once it meets all outstanding technical requirements.

Kind regards,

Chris Connaboy

Academic Editor

PLOS ONE

Additional Editor Comments (optional):

Reviewers' comments:

Reviewer's Responses to Questions

**Comments to the Author**

1. If the authors have adequately addressed your comments raised in a previous round of review and you feel that this manuscript is now acceptable for publication, you may indicate that here to bypass the “Comments to the Author” section, enter your conflict of interest statement in the “Confidential to Editor” section, and submit your "Accept" recommendation.

Reviewer #1: All comments have been addressed

Reviewer #2: All comments have been addressed

2. Is the manuscript technically sound, and do the data support the conclusions?

Reviewer #1: Yes

Reviewer #2: Yes

3. Has the statistical analysis been performed appropriately and rigorously? 

Reviewer #1: Yes

Reviewer #2: Yes

4. Have the authors made all data underlying the findings in their manuscript fully available?

Reviewer #1: Yes

Reviewer #2: Yes

5. Is the manuscript presented in an intelligible fashion and written in standard English?

Reviewer #1: Yes

Reviewer #2: Yes

6. Review Comments to the Author

Reviewer #1: (No Response)

Reviewer #2: Thank you for the well written and considered comments. I think this is a much improved version, and I am happy to recommend this for publication.

7. PLOS authors have the option to publish the peer review history of their article (what does this mean?). If published, this will include your full peer review and any attached files.

Reviewer #1: No

Reviewer #2: No

---

## [Editor Report · Acceptance letter]

23 Nov 2022

PONE-D-22-19072R1 

Evaluating the influence of a constraint manipulation on technical, tactical and physical athlete behaviour 

Dear Dr. Teune:

I'm pleased to inform you that your manuscript has been deemed suitable for publication in PLOS ONE. Congratulations! Your manuscript is now with our production department. 

Kind regards, 

on behalf of

Dr. Chris Connaboy 

Academic Editor

PLOS ONE